# Levosimendan Ameliorates Cardiopulmonary Function but Not Inflammatory Response in a Dual Model of Experimental ARDS

**DOI:** 10.3390/biomedicines10051031

**Published:** 2022-04-29

**Authors:** René Rissel, Moritz Gosling, Jens Kamuf, Miriam Renz, Robert Ruemmler, Alexander Ziebart, Erik K. Hartmann

**Affiliations:** Department of Anaesthesiology, Medical Center of the Johannes Gutenberg-University, Langenbeckstr. 1, 55131 Mainz, Germany; moritz.gosling@gmail.com (M.G.); kamuf@uni-mainz.de (J.K.); miriam.renz@unimedizin-mainz.de (M.R.); robert.ruemmler@email.de (R.R.); alexander.ziebart@unimedizin-mainz.de (A.Z.); hartmane@uni-mainz.de (E.K.H.)

**Keywords:** ARDS, intensive medicine, levosimendan, milrinone, animal model

## Abstract

The calcium sensitiser levosimendan, which is used as an inodilator to treat decompensated heart failure, may also exhibit anti-inflammatory properties. We examined whether treatment with levosimendan improves cardiopulmonary function and is substantially beneficial to the inflammatory response in acute respiratory response syndrome (ARDS). Levosimendan was administered intravenously in a new experimental porcine model of ARDS. For comparison, we used milrinone, another well-known inotropic agent. Our results demonstrated that levosimendan intravenously improved hemodynamics and lung function in a porcine ARDS model. Significant beneficial alterations in the inflammatory response and lung injury were not detected.

## 1. Introduction

Acute respiratory distress syndrome (ARDS) is an intensive care disease, which is associated with a mortality of around 40% due to its possible fulminant course [1]. Pneumonia and sepsis are considered common causes of ARDS. Pathophysiologically, ARDS is a generalised inflammatory reaction of the lungs [2], which leads to damage to the alveolocapillary unit and consecutively to massive permeability oedema. These pathological changes result in ARDS-typical changes, such as the development of atelectasis and dystelectasis, interalveolar oedema, the formation of a fibrotic thickened alveolocapillary membrane, and an increase in the intrapulmonary shunt [2]. The result is a decrease in pulmonary compliance, and consecutive gas exchange impairment occurs. Due to the pronounced hypoxaemia, invasive lung-protective ventilation is often required [3]. Moderate hypercapnia can be tolerated. Supportive therapy regimes, such as the prone position, are associated with improved oxygenation and survival [2]. Due to the pathophysiology of pulmonary inflammation, various drug approaches for the treatment of ARDS were investigated [4,5], but none of the approaches examined demonstrated beneficial results. In particular, protection of the right heart to prevent pulmonary hypertension plays a key role in supportive therapy regimes and influences outcomes [6].

A new approach for affecting immunomodulation within ARDS is inhaled levosimendan [7]. Levosimendan is approved for the treatment of acute decompensated chronic heart failure. First, levosimendan affects calcium sensitivity of the contractile apparatus of cardiac troponin C and improves inotropy without interfering with the calcium metabolism or increasing myocardial oxygen consumption. Second, levosimendan interacts with ATP-dependent potassium channels in smooth muscle cells and causes peripheral vasodilation with a corresponding reduction in the afterload [8]. The first results showed that inhaled levosimendan reduced the release of pro-inflammatory cytokines in a rodent sepsis model [7]. Likewise, the prophylactic inhalation of levosimendan reduced the release of inflammatory mediators and improved survival in a ventilator-induced lung injury (VILI) model [9]. Hence, beyond the haemodynamic aspects, a direct anti-inflammatory effect of levosimendan seems possible. Reducing the inflammatory response and improving right cardiac function are two major issues for patient survival in ARDS. Therefore, we sought to further explore these observations in a translational large animal model. We hypothesised that in a porcine model of severe ARDS levosimendan would (1) intravenously improve hemodynamic and pulmonary function, and (2) mitigate the inflammatory response and lung injury directly and not because of pure haemodynamic stabilisation. For comparison, a control/vehicle group was applied as well as milrinone, a phosphodiesterase-3 (PDE-3) inhibitor, to account for the inodilation-related haemodynamic effects with an alternative mechanism.

## 2. Materials and Methods

After approval by the institutional and state animal care committees (Landesuntersuchungsamt Rheinland-Pfalz, Koblenz, Germany; approval number G18-1-044), we performed this prospective randomised animal study in accordance with the international guidelines for the care and use of laboratory animals [10].

### 2.1. Anaesthesia and Instrumentation

Twenty healthy male pigs (sus scrofa domestica, mean weight: 31.2 ± 1.2 kg) were sedated with an intramuscular injection of azaperon (4 mg kg^−1^) and ketamine (2 mg kg^−1^) and delivered by a local breeder. An ear vein was cannulated and general anaesthesia was induced and maintained by intravenous administration of propofol (Fresenius Kabi, Bad Homburg, Germany; 4 mg kg^−1^ followed by 8–12 mg kg^−1^ h^−1^) and fentanyl (Janssen-Cilag, Neuss, Germany; 4 μg kg^−1^ followed by 0.1–0.2 mg h^−1^). A single dose of atracurium (HEXAL AG, Holzkirchen, Germany; 0.5 mg kg^−1^) was applied, and endotracheal intubation was performed. Pressure-controlled ventilation was initiated: tidal volume 6 mL kg^−1^, positive end-expiratory pressure (PEEP) 5 cm H_2_O, an inspiration to expiration ratio of 1:2, fraction of inspired oxygen (FiO_2_) 0.4 and respiratory rate between 25 and 35 per minute to achieve normocapnia. Haemodynamic monitoring was established with an ultrasound-guided Seldinger technique: a pulmonary artery catheter, an arterial line for blood pressure monitoring and repetitive blood gas analysis, a central venous line and a pulse contour cardiac output catheter (PiCCO, Pulsion Medical, Munich, Germany) were placed via the femoral vessels, as previously described [11]. All the animals received a background infusion of balanced electrolyte solution (Sterofundin, B. Braun, Melsungen, Germany, 5 mL kg^−1^ h^−1^). All haemodynamic and ventilation data were continuously recorded (Datex S/5, GE Healthcare, Solingen, Germany). Normothermia was maintained by body surface warming.

### 2.2. Extended Respiratory Monitoring

The end-expiratory lung volume was determined semi-automatically through the Engström Carestation using the nitrogen wash-out/wash-in method with a FiO_2_ change of 0.1 as described by Olegard and co-workers [12]. The development of pulmonary oedema was assessed by the transpulmonary thermodilution-derived extravascular lung water index (EVLWI (ml kg^−1^); PiCCO, Pulsion Medical, Munich, Germany) and the post-mortem wet/dry ratio (W/D). The alveolar fluid clearance (AFC (%)) was calculated as reported by Hartmann et al., using the amount of non-recovered lavage fluid as the reference: (EVLWBaseline (mL)–EVLW8h (mL)) × 100/non-recovered lavage fluid (mL) [13]. To analyse the regional ventilation distribution, we used an electrical impedance tomography device (Goe-MF II, CareFusion, San Diego, CA, USA) that records thoracic impedance variations associated with tidal ventilation. The electrodes were placed on a transverse lung section just below the axilla. The regional ventilation distribution was examined for the non-dependent, central and dependent lung regions (named levels L1–L3) as a percentage of the global tidal amplitude [14].

### 2.3. ARDS Induction

Following instrumentation, the baseline parameters were assessed at a healthy state. Afterward, ARDS was induced by a double-hit [11]. First, repeated bronchoalveolar lavage over the endotracheal tube with a 30 mL kg^−1^ heated isotonic solution was performed. The endotracheal tube was clamped in inspiration, and the lavage set was connected and immediately instilled and drained by gravity. The lavage procedures were repeated until a ratio of arterial partial pressure of oxygen (PaO_2_) and FiO_2_ ≤ 250 mmHg was achieved. The amount of instilled and drained fluid was recorded. Second, oleic acid (0.1 mL kg^−1^) was given intravenously. The oleic acid (Ölsäure, Applichem GmbH Darmstadt, Germany) was dissolved in a balanced electrolyte solution in a ratio of 1:10 and then applied in fractions of 1–2 mL over 30 min. Short-term hemodynamic instability, which regularly occurs immediately after injection, was treated by norepinephrine boli of 5–10 μg. The procedure was continued until the quotient of arterial PaO_2_ and FiO_2_ was <100 mmHg over 15 min or until a dose maximum of 0.3 mL kg^−1^ was administered. This was followed by a stabilisation phase of 15 min. Afterward, all parameters for the time point T0 were assessed.

### 2.4. Study Protocol and Measured Parameters

After ARDS induction and stabilisation, randomisation of the study medication to one of three groups was performed by blinded observers using a random generator, same-coloured perfusion syringes and from the study assistant programmed hidden running rates.


Levosimendan (Simdax, Orion Pharma, Espoo, Finland; Bolus 24 µg kg^−1^ over 20 min, followed by 0.3 µg kg^−1^ min) intravenous;Milrinone (Sanofi Aventis GmbH, Wien, Austria; Bolus 25 µg kg^−1^ over 10 min, followed by 0.6 µg kg^−1^ min) intravenous;Vehicle group (Glucose 5%, B. Braun Melsungen AG, Melsungen, Germany), 5.0 mL/h intravenous.


Figure 1 summarises the experimental protocol.

The animals were monitored for eight hours after ARDS induction. The ventilation and extended hemodynamic parameters were recorded continuously (Datex S/5, GE Healthcare, Solingen, Germany). Blood gas samples (Rapidlab 248, Bayer Healthcare, Oststeinbek, Germany) were collected every hour to adjust the ventilation, based on the ARDS low PEEP table to achieve a PaO_2_ between 60–120 mmHg and peripheral capillary oxygen saturation (SpO_2_) above 92%. To maintain hemodynamic stability (mean arterial pressure > 60 mmHg) and to avoid instability, the animals were treated by continuous central venous noradrenaline infusion. The hematological parameters were determined at baseline, T4, and T8. The plasma levels of TNF-alpha, interleukin-6 (IL-6), and the soluble receptor for advanced glycation (sRAGE) were determined by quantifying enzyme-linked immunosorbent assays (Porcine IL-6 Quantikine ELISA, Porcine TNF-alpha Quantikine ELISA, Porcine sRAGE Quantikine ELISA, R&D Systems, Wiesbaden, Germany) according to the manufacturer’s instructions. At the end of the experiment, the animals were euthanised under general anesthesia by central venous injection of propofol (200 mg) and potassium chloride (40 mval).

### 2.5. Histopathological Parameters

After the experiment, the lungs were ventilated with the previous ventilatory settings until they were removed en bloc from the chest. A bloating manoeuvre was omitted. The post-mortem pulmonary expressions of inflammatory markers (IL-6, IL-1b, COX-2, iNOS, and TNF-alpha) were determined in cryopreserved lung samples from the right lower lobe for mRNA analysis by real-time polymerase chain reaction (rt-PCR; Lightcycler 480 PCR System; Roche Applied Science, Penzberg, Germany.) The mRNA expression was normalised to peptidylprolyl isomerase A. The right ventral upper and lower lobes were evaluated for histological evidence of pulmonary damage. Hence, the lung samples were fixed in formalin for paraffin embedding and stained using haematoxylin/eosin dye. To determine the lung damage, we used a standardised scoring system, as described in detail by our group previously [14]. The exsanguinated left lung was weighed, sliced, and dried for determination of the wet-to-dry ratio.

### 2.6. Statistics

All parameters are presented as mean and standard deviation (±SD) or displayed as error bars and multiple scatter lines. The analysis focuses on the relevant time points at baseline, T0, T4, and T8. The group effects over time (levosimendan vs. milrinone vs. vehicle) were compared by two-way analysis of variance (ANOVA) with a post-hoc Student–Newman–Keuls test. The secondary outcome parameters (i.e., wet-to-dry/ratio and AFC) were compared by the Kruskall–Wallis test. A *p*-value lower than 0.05 was accepted as significant. The software package SigmaPlot 12.5 (Systat Software, San Jose, CA, USA) was used.

## 3. Results

The study protocol was completed in all 20 animals (weight: 31.2 ± 1.2 kg; levosimendan *n* = 8, milrinone *n* = 8, vehicle *n* = 4). During the ARDS induction, no adverse events occurred. The hemodynamic and respiratory parameters did not differ at baseline. After ARDS induction, the PaO_2_/FiO_2_ ratio significantly decreased in all groups at T0 (*p* < 0.05). Afterward, the PaO_2_/FiO_2_ ratio significantly increased in the animals treated with levosimendan (*p* < 0.05 vs. milrinone/vehicle; Figure 2). There is a rapid decrease of the effect, similar to that seen with milrinone or vehicle.

Further, the minute ventilation showed a steady significant rise in all groups over time (*p* < 0.05; Table 1). At T8, the animals in the vehicle group required higher minute ventilation compared to the levosimendan group (*p* < 0.031; Table 1). The functional residual capacity decreased significantly after the ARDS induction compared to the baseline in all groups without any further intergroup differences over eight hours (*p* < 0.05; Table 1). The EVLWI remained significantly elevated after ARDS induction in all groups over the experiment’s duration without any intergroup differences (*p* < 0.05; Table 2). All other respiratory parameters showed no relevant alterations (Table 1).

A significant increase in the mean pulmonary arterial pressure following ARDS induction was observed in all groups compared to the baseline (*p* < 0.05; Figure 3). The mean pulmonary arterial pressure remained significantly lower in the levosimendan group for the rest of the experiment (*p* < 0.001 levosimendan vs. milrinone; *p* < 0.05 levosimendan vs. vehicle; Figure 3).

Further hemodynamic parameters showed no relevant differences (Table 2). The concentration of C-reactive protein increased noticeably more in the vehicle and milrinone groups compared to the levosimendan group without reaching a level of significance (Table 3).

The blood gas analyses showed no relevant differences (Table 4). At baseline, the inflammatory markers in the plasma showed no differences. Levosimendan tended to suppress the sRAGE and IL-6 plasma levels more than milrinone or vehicle over eight hours (*p* = 0.055 for intergroup comparisons; Figure 4). The IL-6 levels increased significantly in all groups compared to the baseline levels (Figure 4).

The pulmonary tissue mRNA expression revealed a significant upregulation of COX-2 (*p* < 0.05 levosimendan vs. milrinone; Figure 5).

The regional distribution of ventilation did not differ between the groups. Lung injury scoring showed no relevant intergroup differences. The alveolar fluid clearance showed no significant changes. The tissue wet-to-dry ratio as a surrogate of oedema formation did not differ between the groups (*p* > 0.05; levosimendan 6.3 ± 1.7; milrinone 7.5 ± 1.4; vehicle 6.7 ± 1.3).

## 4. Discussion

In this study, we investigated the effects of levosimendan administered intravenously on hemodynamics and pulmonary function after experimental ARDS induction. Further, we examined the inflammatory response and lung injury.

The chosen new double-hit ARDS model is highly reproducible and suitable [11]. Combining the oleic acid infusion and bronchoalveolar lavage mimics the main pathophysiological changes seen in human ARDS [11], resulting in a severe and persistent impaired gas exchange over eight hours. As reported, mean pulmonary arterial pressure initially increased due to ARDS. This common phenomenon with right heart strain and failure is similar to human ARDS [6,15].

Levosimendan, an inotropic agent, is effective for hemodynamic stabilisation of patients with acute heart failure [16]. Levosimendan improves inotropy and causes peripheral vasodilation with a corresponding reduction in afterload [8]. Several studies have indicated that levosimendan administration has immunomodulatory effects, even when administered by inhalation [9,17,18]. Most studies administer the drug as preconditioning, which is unrealistic in clinical practice, as levosimendan is not indicated in the early disease stage [19]. Furthermore, in pulmonary edema, pulmonary hypertension, and atelectasis, it remains unclear what proportion of the drug reaches the alveolar space and lung tissue when inhaled [9]. For this reason, we administered levosimendan intravenously in early ARDS. Once administered, we observed an initial decrease in cardiac output followed by a steady increase. Further, no cardiac output undulations, as seen in the milrinone group, were recorded. Theoretically, cardiac output improvement contributes to the downregulation of inflammatory transcriptional factors such as NF-κB [20]. As a result, the production of cytokines is inhibited. Further, activation of the ATP-dependent potassium channels in the mitochondrion of damaged endothelial cells attenuates the activity of NF-κB [21]. Attenuated NF-κB activity reduces the expression of tissue factor and plasminogen-activator-inhibitor-1 in damaged endothelial cells [21,22]. This anti-thrombotic and pro-fibrinolytic environment prevents pulmonary microvascular obstruction, a common cause of right heart failure in human ARDS. NF-κB is a key activator of IL-6 and sRAGE production. In our study, IL-6 levels remained lower over the whole trial in the levosimendan group. It is conceivable that levosimendan interacts with the known NF-κB-IL-6 pathway and inhibits the release of pro-inflammatory cytokines. This could also explain the depressed levels of sRAGE. sRAGE is a multiligand cell surface receptor from the immunoglobulin superfamily, with the highest concentrations found in lung tissue [23]. sRAGE plays a key role in ARDS pathology and the development of organ dysfunction [24,25]. Peak concentrations of sRAGE are found in the first days after lung injury. However, it remains unclear if sRAGE is a simple biomarker or a causal factor in lung injury [23]. There is evidence of two activation triggers of sRAGE. First, injury to lung parenchyma caused distortion of the alveolar cells as part of a VILI [26]. In the present study, this part of the activation was excluded under consideration of lung-protective ventilation without intergroup differences. Second, sRAGE can be produced from alveolar type I cells after inflammatory activation of the NF-κB pathway [27,28]. This is also of concern, considering IL-6 as another important biomolecule in ARDS pathology. IL-6 plays a key role at the onset and progression of ARDS and can contribute to multiple organ dysfunction syndrome [29,30]. In recent years, IL-6 has become a target of interest for clinical intervention due to its context-dependent pro- and anti-inflammatory properties [31]. The pathway for controlling IL-6 activity is difficult and complex. In many diseases, even in new-onset COVID-19 ARDS, persistent elevation of IL-6 predicts a poor outcome [31,32]. One key activator of the IL-6 pathway is based on the upregulation of NF-κB, as already described above for sRAGE activation [33].

The calcium-sensitising effect of levosimendan reduces the intracellular concentration of free calcium molecules. It is reported that calcium overload in heart cells induces apoptotic and inflammatory pathways [34]. This correlates with decreased heart function. Further, levosimendan protects endothelial cells from death. In human heart microvascular endothelial cells treated with levosimendan, the intercellular adhesion molecule 1 (ICAM-1) is almost entirely suppressed [21]. ICAM-1 is responsible for the binding and extravasation of granulocytes, which result in the formation of pulmonary oedema and local inflammation. In our study, the wet-to-dry ratio and the reinforced alveolar fluid clearance capacity as surrogate parameters of local edema and inflammation showed no beneficial effects in the levosimendan group.

Milrinone is a phosphodiesterase-3 inhibitor that enhances cardiac contractility by increasing the levels of intracellular cyclic adenosine monophosphate (cAMP) [35]. cAMP was considered to have general anti-inflammatory properties [36,37]. Further, milrinone modulates and attenuates the systemic release of cytokines after cardiopulmonary bypass within 24 h [35]. Less evidence and conflicting reports exist compared to the anti-inflammatory effects of levosimendan. The effects of the two inotropic agents on cardiac inflammation and left ventricular performance were examined in mice with caecal ligation and puncture-induced sepsis [38]. Cardiac inflammation decreased in the levosimendan group and increased in the milrinone group. Further, inotropy was impaired in the mice when treated with levosimendan and preserved in the milrinone group. Both agents alleviated cardiac injury. Similar findings were detected in our study (i.e., reduced systemic inflammation and improved cardiopulmonary function), thus supporting the different advantages of levosimendan. However, it is critical to note that it is unclear to what extent levosimendan or its metabolites are involved in these positive phenomena. The half-life of levosimendan is one hour [39]. OR-1896 is one active metabolite of levosimendan with a half-life of approximately 70 h [39]. Different previous studies reported similar hemodynamic and inflammatory effects, even after a single bolus of levosimendan [7]. We may hypothesise that OR-1896 could also potentially account for the effectiveness after the infusion of levosimendan as observed in our trial. Actually, there are no data in the literature supporting this theory.

The pulmonary tissue markers of inflammation were elevated or preserved in the animals treated with levosimendan. In particular, cyclooxygenase-2 (COX-2) was significantly elevated compared to the milrinone group. COX-2 is an enzyme from the arachidonic acid metabolic pathway and produces different prostaglandins. Prostaglandins sustain homeostatic functions and mediate pathogenic mechanisms, especially inflammatory and anti-inflammatory responses [40]. COX-2 is highly expressed in the airway epithelium [41]. It is well known that COX-2 is upregulated early (at 2 h) at sites of acute inflammation [42]. These anti-inflammatory responsive elements were shown to regulate the expression of detoxification and antioxidant enzymes via NF-E2–related factor 2 (Nrf2) [42]. In mouse models, Nrf2 was shown to play an essential role in protection against acute lung inflammation and damage induced by a number of stimuli [43]. It was demonstrated that prostaglandins from COX-2 directly activate Nrf2 [44]. Cytokines and hypoxia are known to stimulate COX-2 expression [45,46]. Interactions with levosimendan or milrinone have not yet been described. It remains unclear if the COX-2 expression in our study was significantly elevated due to the aforementioned anti-inflammatory response effects or despite direct interactions with levosimendan. Further investigations are needed to understand this phenomenon.

Our study has some limitations. It was difficult to determine the extent to which lung injury is caused by lavage, oleic acid, or both. This is common to other combined animal models. Therefore, it is important to have a study protocol with predefined target values (e.g., PaO_2_/FiO_2_ ratio). Dysregulated inflammation, as measured in our study, is widely known as a major part of the progress of various diseases. The significance of changes in pro- and anti-inflammatory cytokines in animals remains unclear when translated to humans. Further human studies will be necessary to clarify this. Due to local conditions, we were not allowed to extend the experiment’s duration. The study therefore only represents pathological changes in early-stage ARDS. Late changes in the pathology were not recorded. Many hospital patients will not be seen in such an early stage of the disease. Further studies will be needed to analyse late-stage ARDS. To distinguish between the effects of levosimendan or its metabolite OR-1896, the measurement of the effectiveness of plasma concentrations of both compounds should be performed.

## 5. Conclusions

To summarise the present study and in reference to our hypothesis, we demonstrated that (1) levosimendan intravenously improved hemodynamics and lung function in early porcine ARDS. Thereby, (2) significant beneficial alterations in the inflammatory response and lung injury were not fully conclusive. The potential beneficial immunomodulatory effects of levosimendan in the ARDS need to be investigated more closely.

## Figures and Tables

**Figure 1 biomedicines-10-01031-f001:**
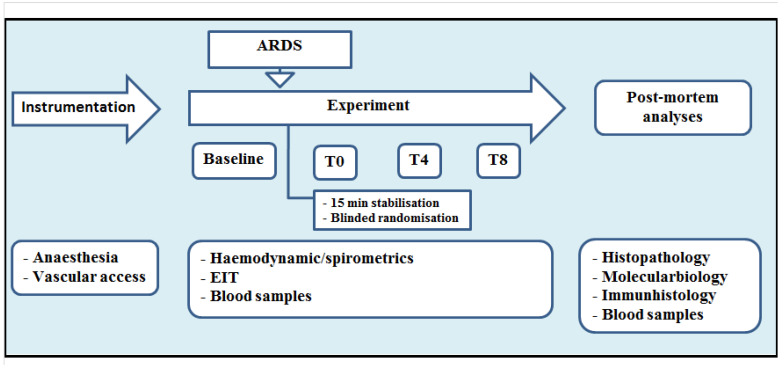
Experimental flowchart. EIT: electrical impedance tomography. ARDS: acute respiratory distress syndrome. T_x_: timepoint.

**Figure 2 biomedicines-10-01031-f002:**
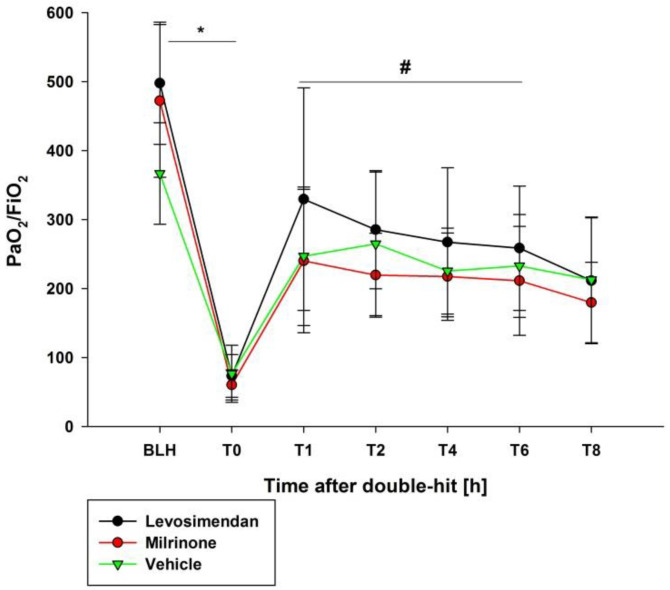
PaO_2_/FiO_2_ ratio. * indicates *p* < 0.05 BLH vs. T0 for all groups. # indicates *p* < 0.05 levosimendan vs. milrinone T1–T6. h: hours. PaO_2_: partial pressure of oxygen. FiO_2_: fraction of inspired oxygen. BLH: baseline healthy. T_x_: timepoint.

**Figure 3 biomedicines-10-01031-f003:**
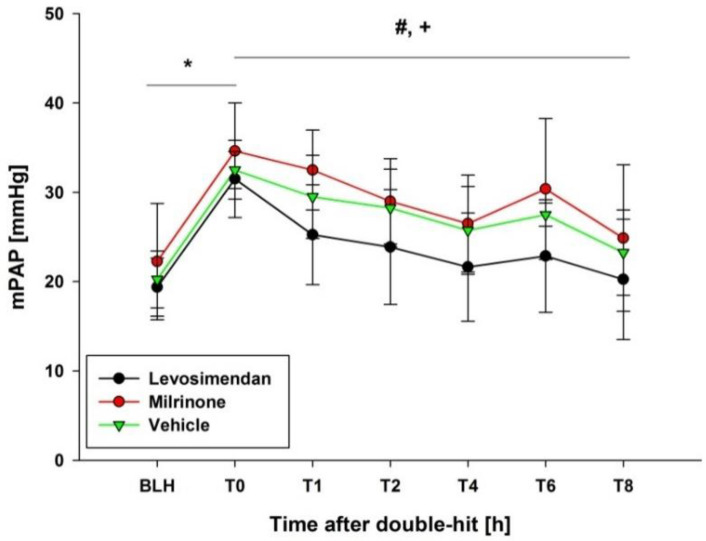
mPAP. * indicates *p* < 0.05 BLH vs. T0 for all groups. # indicates *p* < 0.001 levosimendan vs. milrinone. + indicates *p* < 0.05 levosimendan vs. vehicle. mPAP: mean pulmonary arterial pressure. BLH: baseline healthy. Tx: timepoint. h: hours.

**Figure 4 biomedicines-10-01031-f004:**
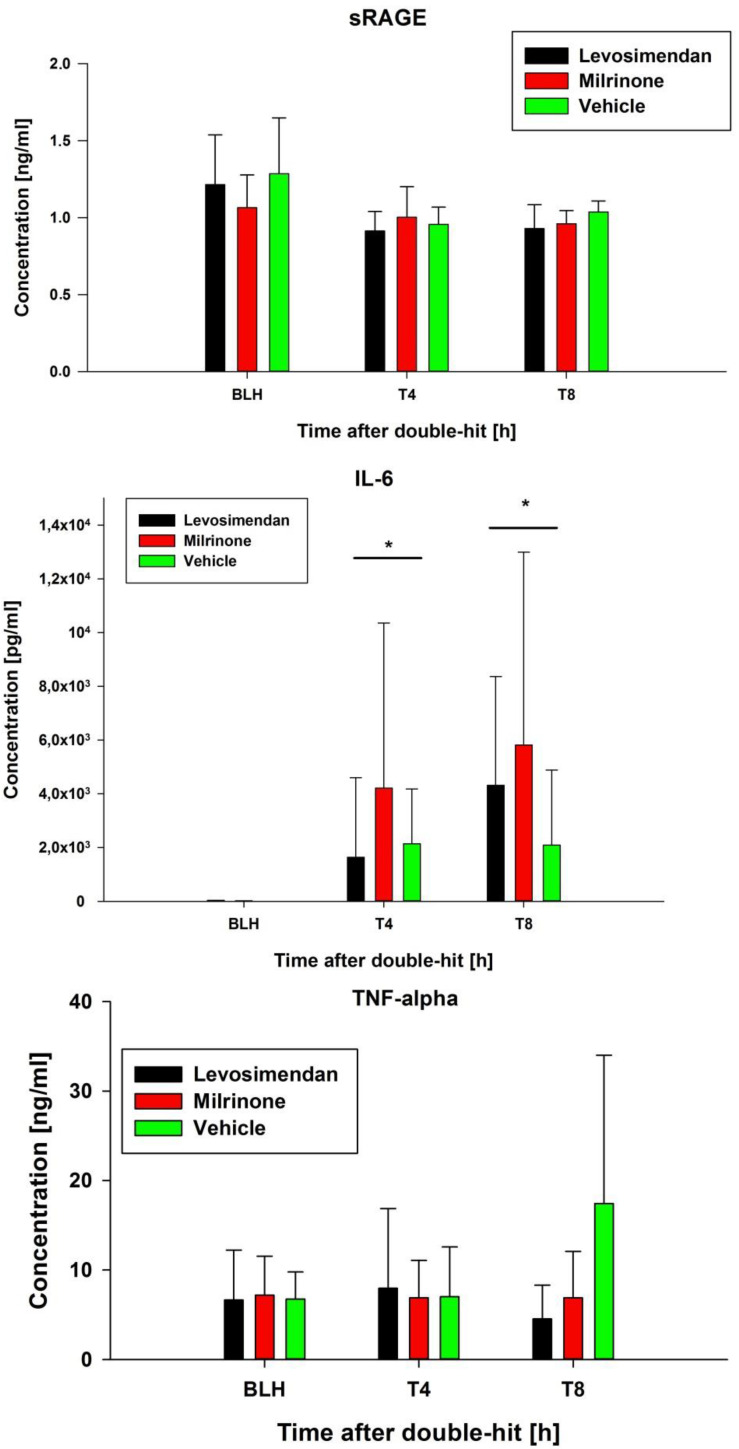
Blood sample analysis of inflammatory markers. * indicates *p* < 0.01 sham/milrinone/levosimendan for IL-6 for T4/T8 vs. BLH. sRAGE: soluble receptor for advanced glycation; IL-6: Interleukin-6. BLH: baseline healthy. Tx: timepoint.

**Figure 5 biomedicines-10-01031-f005:**
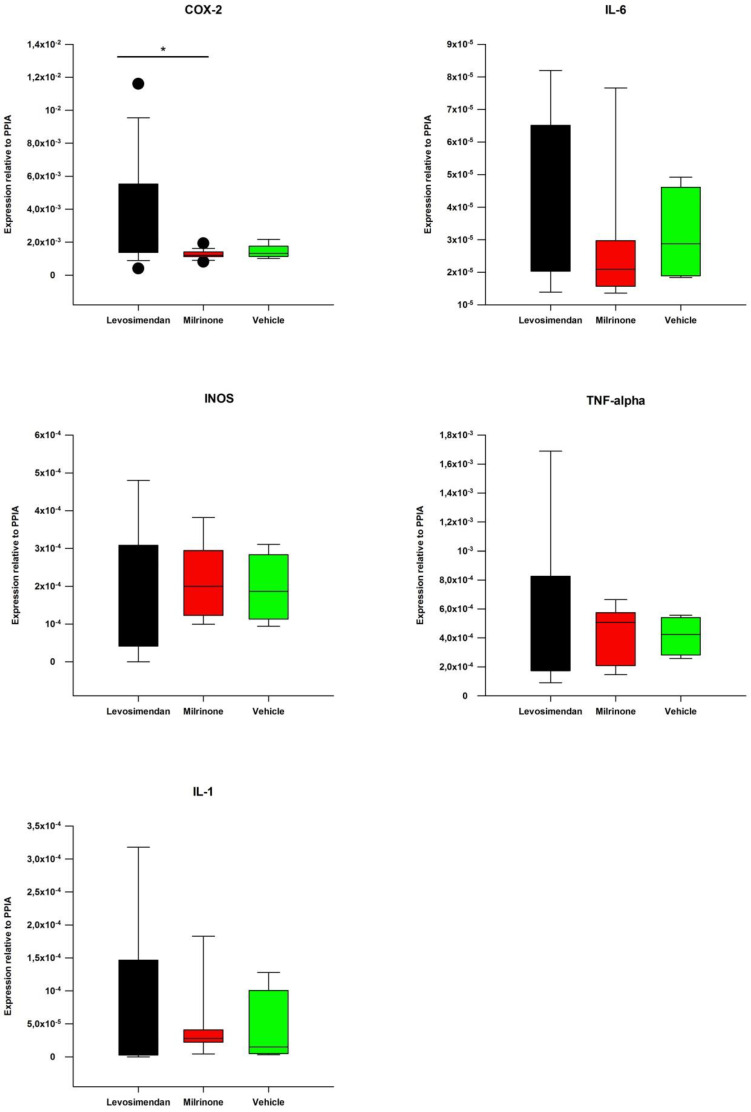
Tissue sample analyses. Inflammatory marker expressions of COX-2, IL-6, IL-1b, INOS, TNF-alpha. * indicates *p* < 0.05 levosimendan vs. milrinone for COX-2. • statistical outliers. COX-2: Cyclooxygenase-2. IL-6: Interleukin-6. IL-1b: Interleukin-1b. INOS: Inducible nitric oxide synthase.

**Table 1 biomedicines-10-01031-t001:** Respiratory data over time. Values are mean (SD). * indicates *p* < 0.05 vs. baseline value. # indicates *p* < 0.05 in intergroup comparison. SpO_2_: oxygen saturation; AaDO_2_: alveolo–arterial oxygen difference; PaO_2_: arterial oxygen; FiO_2_: fraction of inspired oxygen; PaO_2_/FiO_2_: oxygen index; FRC: fraction of inspired oxygen; MV: minute volume; TV: tidal volume; Ppeak: peak inspiratory pressure; Pmean: mean airway pressure; PEEP: positive end-expiratory pressure. BLH: baseline healthy.

Parameter	Group	BLH	T0	T4	T8
		MEAN (SD)	MEAN (SD)	MEAN (SD)	MEAN (SD)
	Levosimendan	99 (1)	89 (6)	96 (3)	95 (5)
SpO_2_	Milrinone	98 (3)	76 (17)	98 (2)	97 (1)
(%)	Vehicle	96 (3)	84 (6)	97 (1)	96 (1)
	Levosimendan	0.4 (0.0)	1.0 (0.0)	0.55 (0.1)	0.5 (0.1)
FiO_2_	Milrinone	0.4 (0.0)	1.0 (0.0)	0.54 (0.1)	0.49 (0.1)
(%)	Vehicle	0.4 (0.0)	1.0 (0.0.)	0.53 (0.1)	0.5 (0.1)
	Levosimendan	498 (88) #	73 (31) #/*	267 (108) #	211 (91)
PaO_2_/FiO_2_	Milrinone	472 (110)	60 (22) *	217 (63)	180 (58)
(mmHg)	Vehicle	367 (74)	76 (41) *	225 (62)	212 (91)
	Levosimendan	34 (33)	572 (28)	197 (80)	203 (108)
AaDO_2_	Milrinone	47 (39)	546 (55)	218 (77)	225 (72)
(mmHg)	Vehicle	85 (29)	487 (204)	201 (53)	197 (61)
	Levosimendan	637 (157)	265 (79) *	404 (126)	413 (91)
FRC	Milrinone	539 (126)	259 (262) *	393 (87)	356 (130)
(ml)	Vehicle	535 (63)	222 (92) *	378 (73)	375 (70)
	Levosimendan	5.7 (1.1)	5.7 (0.6)	6.7 (0.5) *	6.6 (0.5) *
MV	Milrinone	5.8 (1.0)	5.4 (1.3)	7.3 (1.0) *	7.5 (1.0) *
(l min^−1^)	Vehicle	6.7 (0.6)	7.1 (0.5) #	7.8 (1.4)	8.1 (1.4) */#
	Levosimendan	5.5 (0.4)	5.6 (0.5)	5.5 (0.4)	5.4 (0.4)
TV	Milrinone	5.7 (0.6)	5.2 (1.4)	5.9 (0.6)	5.9 (0.4)
(ml kg^−1^)	Vehicle	6.0 (0.4)	6.2 (0.2)	6.0 (0.3)	6.1 (0.3)
	Levosimendan	15 (2)	25 (5)	25 (5)	24 (6)
Ppeak	Milrinone	16 (2)	30 (7)	26 (3)	25 (3)
(mbar)	Vehicle	15 (1)	28 (5)	25 (4)	24 (4)
	Levosimendan	7.7 (0.4)	11.1 (1.8)	14.1 (3.2)	13.2 (3.4)
Pmean	Milrinone	8.0 (0.5)	15.6 (6.0)	14.5 (2.1)	12.7 (2.3)
(mbar)	Vehicle	8.0 (0.0)	13.5 (2.6)	13.5 (3.1)	12.2 (3.2)
	Levosimendan	5 (0)	10 (3)	9 (3)	8 (2)
PEEP	Milrinone	5 (0)	10 (6)	9 (2)	7 (2)
(cm H_2_O)	Vehicle	5 (0)	10 (2)	7 (3)	7 (2)

**Table 2 biomedicines-10-01031-t002:** Hemodynamic data over time. Values are mean (SD). * indicates *p* < 0.05 vs. baseline value. # indicates *p* < 0.05 in intergroup comparison. MAP: mean arterial pressure; HR: heart rate; mPAP: mean arterial pulmonary pressure; CO: cardiac output; PCWP: pulmonary capillary wedge pressure; GEDVI: global endiastolic volumen index; EVLWI: end-diastolic lung water index; CVP: central venous pressure; ScO_2_: cerebral oxygen saturation. BLH: baseline healthy.

Parameter	Group	BLH	T0	T4	T8
		MEAN (SD)	MEAN (SD)	MEAN (SD)	MEAN (SD)
	Levosimendan	67 (6)	74 (11)	70 (10)	64 (10)
MAP	Milrinone	67 (8)	73 (14)	69 (7)	66 (6)
(mmHg)	Vehicle	73 (4)	78 (8)	70 (9)	70 (7)
	Levosimendan	79 (14)	110 (32)	119 (43) *	112 (46)
HR	Milrinone	80 (14)	113 (30)	125 (44)	130 (44) *
(min^−1^)	Vehicle	101 (16)	104 (33)	119 (22)	103 (21)
	Levosimendan	19 (3)	31 (4) #/*	21 (6) #	20 (6) #
mPAP	Milrinone	22 (6)	34 (5) #/*	26 (5) #	24 (8) #
(mmHg)	Vehicle	20 (3)	32 (2) *	25 (4)	23 (4)
	Levosimendan	7 (3)	6 (3)	6 (3)	6 (3)
CVP	Milrinone	7 (3)	7 (2)	7 (2)	7 (2)
(mmHg)	Vehicle	6 (1)	5 (2)	5 (1)	5 (1)
	Levosimendan	3.3 (0.7)	3.0 (0.8)	3.3 (1.2)	3.7 (1.3)
CO	Milrinone	3.4 (0.9)	3.7 (0.9)	3.2 (0.5)	4.3 (0.8)
(l min^−1^)	Vehicle	4.6 (0.8)	3.9 (1.0)	3.2 (0.2)	3.5 (0.2)
	Levosimendan	9 (3)	8 (2)	8 (2)	8 (4)
PCWP	Milrinone	10 (2)	10 (2)	10 (1)	9 (2)
(mmHg)	Vehicle	9 (1)	9 (1)	8 (1)	8 (1)
	Levosimendan	425 (117)	440 (149)	409 (87)	439 (104)
GEDVI	Milrinone	468 (168)	453 (79)	399 (55)	432 (66)
(mL m^−2^)	Vehicle	464 (108)	484 (115)	400 (45)	418 (68)
	Levosimendan	10.3 (1.3)	20.8 (7.4) *	18.3 (3.6) *	18.5 (4.9) *
EVLWI	Milrinone	11.5 (2.5)	23.1 (5.5) *	18.2 (3.5) *	18.2 (3.7) *
(mL kg^−1^)	Vehicle	11.2 (0.9)	21.2 (5.7) *	18.0 (3.1) *	16.7 (4.5) *
	Levosimendan	51 (8)	36 (10)	53 (8)	58 (4)
ScO_2_	Milrinone	50 (11)	32 (12)	54 (11)	60 (10)
(%)	Vehicle	48 (6)	35 (5)	53 (9)	51 (2)

**Table 3 biomedicines-10-01031-t003:** Laboratory diagnostics. Values are mean (SD). BLH: baseline healthy; CRP: C-reactive protein.

Parameter	Group	BLH	T4	T8
		MEAN (SD)	MEAN (SD)	MEAN (SD)
	Levosimendan	10.5 (5.9)	14.3 (13.4)	12.7 (6.5)
Leucocytes	Milrinone	12.5 (4.4)	10.2 (6.4)	12.4 (6.7)
(g/L)	Vehicle	12.9 (5.2)	15.6 (5.7)	12.2 (3.4)
	Levosimendan	8.3 (0.8)	9.4 (1.3)	9.1 (0.6)
Haemoglobin	Milrinone	9.2 (1.1)	10.4 (0.9)	9.1 (0.9)
(g/dL)	Vehicle	9.8 (0.7)	10.8 (1.1)	9.8 (0.8)
	Levosimendan	376 (65)	270 (68)	244 (60)
Thrombocytes	Milrinone	329 (112)	272 (67)	276 (61)
(1000/µL)	Vehicle	395 (76)	303 (92)	295 (79)
	Levosimendan	0.26 (0.1)	0.31 (0.18)	0.34 (0.16)
CRP	Milrinone	0.28 (0.15)	0.38 (0.19)	0.58 (0.16)
(mg dL^−1^)	Vehicle	0.33 (0.15)	0.76 (0.14)	0.91 (0.22)
	Levosimendan	0.82 (0.20)	0.9 (0.14)	0.97 (0.14)
Creatinine	Milrinone	0.81 (0.29)	0.80 (0.20)	0.98 (0.24)
(mg dL^−1^)	Vehicle	0.85 (0.19)	0.9 (0.24)	0.9 (0.23)
	Levosimendan	8 (2)	10 (2)	12 (3)
Urea	Milrinone	5 (3)	8 (2)	11 (1)
(mg dL^−1^)	Vehicle	6 (4)	7 (3)	9 (2)
	Levosimendan	1351 (7230)	1416 (3629)	114 (2286)
Creatine kinase	Milrinone	1257 (2160)	2118 (1217)	1882 (1032)
(U L^−1^)	Vehicle	1468 (1138)	1008 (779)	936 (657)

**Table 4 biomedicines-10-01031-t004:** Blood gas analysis. Values are mean (SD). BE: base excess; PaCO_2_: arterial carbon dioxide; BLH: baseline healthy. # indicates *p* < 0.05 in intergroup comparison.

Parameter	Group	BLH	T0	T4	T8
		MEAN (SD)	MEAN (SD)	MEAN (SD)	MEAN (SD)
	Levosimendan	7.46 (0.06)	7.36 (0.09)	7.47 (0.08)	7.45 (0.09)
pH	Milrinone	7.48 (0.07)	7.28 (0.08)	7.47 (0.06)	7.47 (0.06)
	Vehicle	7.45 (0.03)	7.36 (0.09)	7.48 (0.05)	7.49 (0.04)
	Levosimendan	5.9 (2.8)	3.9 (3.2)	7.2 (3.2) #	6.4 (2.8)
BE	Milrinone	5.5 (2.1)	2.1 (1.7)	5.6 (2.9)	5.9 (2.7)
(mmol mL^−1^)	Vehicle	5.0 (2.6)	2.4 (3.4)	6.8 (2.4)	7.5 (1.5)
	Levosimendan	42 (4)	53 (10)	43 (5)	45 (10)
PaCO_2_	Milrinone	39 (6)	64 (18)	41 (5)	40 (8)
(mmHg)	Vehicle	41 (1)	51 (9)	40 (3)	40 (4)
	Levosimendan	3.5 (0.3)	3.9 (0.5)	4.7 (0.3)	4.5 (0.5)
Potassium	Milrinone	3.4 (0.4)	3.8 (0.4)	4.9 (0.6)	4.4 (0.4)
(mmol L^−1^)	Vehicle	3.4 (0.3)	3.6 (0.2)	4.6 (0.6)	4.3 (0.5)
	Levosimendan	2.2. (1.4)	1.9 (0.7)	1.0 (0.2)	1.8 (2.1)
Lactate	Milrinone	1.6 (1.2)	2.0 (0.8)	1.6 (1.5)	1.4 (1.3)
(mmol L^−1^)	Vehicle	2.2 (2.1)	1.9 (0.4)	0.9 (0.3)	0.7 (0.2)

## Data Availability

Data are available upon reasonable request.

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
