# Peer review of "Levosimendan Ameliorates Cardiopulmonary Function but Not Inflammatory Response in a Dual Model of Experimental ARDS"

_biomedicines, 2022, doi:10.3390/biomedicines10051031_

Round 1

Reviewer 1 Report

No further comments

Author Response

Thanks very much for editing our paper and approving it for publication!

Reviewer 2 Report

The authors performed a prospective randomized study and conducted an experimental porcine acute respiratory distress syndrome (ARDS) model (double-hit ARDS model) to investigate the effect of intravenous levosimendan on cardiopulmonary function and inflammatory response in ARDS. The ventilation, hemodynamic parameters, and blood gas samples were recorded continuously for 8 hours (i.e., the early stage of ARDS). This study found that levosimendan improved hemodynamics and lung function, but the benefit in inflammatory response was not detected.  

The prevalence rate of right ventricular dysfunction by echocardiography in ARDS range from 22% to 50% (CHEST 2017; 152(1):181-193). Severe right ventricular dysfunction was associated with increase mortality. A previous prospective, randomized, placebo-controlled, pilot study had demonstrated that levosimendan improved right ventricular performance in septic patients with ARDS (Crit Care Med 2006; 34:2287–2293). Although this study also found that levosimendan could improve cardiopulmonary function in an experimental ARDS model, the effect on outcome of patients with ARDS still need to be investigated.  

The structure of manuscript is well organized and the discussion was comprehensive. Finally, I had some comments.

Major comments:

  1. The authors used pulse contour cardiac output catheter (PiCCO) for hemodynamic monitor. However, PiCCO device was not suitable under the conditions of right heart failure and severe pulmonary hypertension. Could you explain?
  2. Did the authors analyze the histopathological change of the new double-hit ARDS model, i.e. typical pathological features of ARDS, diffuse alveolar damage?
  3. Did the authors evaluate the right ventricle function or performance by echocardiography in this study?
  4. More than 70% references (42/57) are older than 10 years, and should be updated and revised to smaller numbers.

Minor comments:

  1. Page 1 line 11: inodilatator suggested revised to inodilator
  2. Page 5, line 184: Table 1 should be revised to Table 2 (EVLWI is in Table 2).
  3. The style of the references should be adjusted as MDPI template.

Author Response

Dear reviewer 2, thank you for editing our article! We hope that you are fine with all our changes and answered comments and that you approve our paper for publication, now!

Major comments:

  1. We fully agree that no valid statements can be made about the right heart function with the PiCCO technology. However, PiCCO-derived parameters like cardiac output provide valuable information in ARDS treatment by means of assessing volume status, lung water content or benefits of inotropic agents. Furthermore, it is a relatively less invasive approach and commonly used and validated in pigs. To assess the right heart we used a pulmonary arterial catheter (Swan-Ganz) as described in the paper. PCWP as well as the mPAP measurement were suitable. By combining both methods, we were able to establish comprehensive hemodynamic monitoring and incorporate it into the interpretation of the results. As the following literature recommendations show, the pulmonary arterial catheter is suitable for monitoring right heart function and is widely used in all of our experiments: Krishnan, Anish et al. “Right Heart Catheterisation: How To Do It.” Heart, lung & circulation vol. 28,4 (2019) // Gassanov, Natig, and Fikret Er. [Right heart catheter examination: step by step]. Deutsche medizinische Wochenschrift (1946) vol. 146,16 (2021): 1064-1069.
  2. In all of our experiments, the histological work-up is carried out at the end. A standardized scoring system was used for the evaluation, which was based on the diffuse alveolar damage score (Ziebart A, Hartmann EK, Thomas R, et al. Low tidal volume pressure support versus controlled ventilation in early experimental sepsis in pigs. Respiratory Research 2014; 15: 101 // Spieth PM, Knels L, Kasper M, et al. Effects of vaporized perfluorohexane and partial liquid ventilation on regional distribution of alveolar damage in experimental lung injury. Intensive Care Medicine 2007; 33: 308– 14.) This procedure is highly standardized in our working group. In a not yet published post-hoc analysis, we are currently comparing the damage patterns of a septic ARDS model and the new double-hit model chosen in this study. The double-hit model is particularly convincing and presents the typical pathophysiological changes in the early phase of ARDS. All these findings are in accordance with the statement of the American Thoracic Society and their workshop report to the main features that characterize ARDS in animal models (Matute-Bello, Gustavo et al. “An official American Thoracic Society workshop report: features and measurements of experimental acute lung injury in animals.” American journal of respiratory cell and molecular biology vol. 44,5 (2011): 725-38.)
  3. Unfortunately, we were unable to perform cardiac ultrasound examination in the present experiment due to lack of an echocardiographic probe for use in pigs. We consider the importance of cardiac sonography to be very high and it is carried out daily in our daily clinical routine. Only a linear transducer for vascular puncture was available. Due to Corona funding measures, however, we received an ultrasound probe for cardiac examination at the beginning of the year and will use it in the upcoming studies.
  4. Thank you for the friendly note about the topicality and number of references used. We have tried to update the literature and to reduce the number of references, when applicable.

Minor comments:

  1. Revised to inodilator.
  2. Revised to Table 2.
  3. The style of the references adjusted to the Biomedicines template via Mendely.

Round 2

Reviewer 2 Report

The authors fully responded to my comments and I have no further questions.

This manuscript is a resubmission of an earlier submission. The following is a list of the peer review reports and author responses from that submission.

Round 1

Reviewer 1 Report

The manuscript under review presents data from an animal study assessing the use of intravenous levosimendan in comparison to milirinone and glucose infusion in an ARDS model.

The topic is of interest to the ICU community, the manuscript needs some improvement regarding language and style.

There are 3 main questions that need attention.

  1. Hemodynamic improvement after levosimendan infusion usually becomes obvious after a longer interval. Would you be able to explain the significant changes shown in your results with in a very short time span?
  2. The “blinding process” is unclear. How can the team be blinded if one compound is given over 20 min (clearly levosimendan), another compound is given over 10 min (milrinone) and glucose as a continuous infusion?
  3. Please clarify the time between “stabilization”, “randomization” and T0. Is T0 (in graphs and tables) the same as 0 in the flowchart?

Other:

Line 36: replace “cor pulmonale” by “pulmonary hypertension”

Line 40: replace “sensitizes the calcium sensitivity” by “affects calcium sensitivity”

Line 64: provide reference for international guidelines

Line 114: provide route of administration for all compounds

Line 139: replace “killed” by “euthanized”

Consider using another term for “experiment”, e.g. trial, testing

Line 142: replace “ventilation” by “ventilatory”

Line 169: There is a rapid decrease of the effect, similar as seen with milrione or vehicle

Table 3: Are the units for leucocytes, hemoglobin and platelets “%”?

Reviewer 2 Report

In the present study, Rissel and colleagues evaluated the effect of levosimendan in a porcine ARDS model. They induced ARDS in healthy male pigs by combining oleic acid infusion and bronchoalveolar lavage, and examined the clinical outcomes by intravenous levosimendan administration. They found that levosimendan treatment improved haemodynamics and lung function, but not the inflammatory response during early porcine ARDS. The study employed a double-hit large animal ARDS model, which provides a superior clinical relevance to human ARDS compared to regular rodent ARDS models. However, there are several major issues needing to be addressed:

  1. The reviewer could not access to the Supplemental information to evaluate the established porcine ARDS model.
  2. Several previous reports have demonstrated the immunoregulatory effect of levosimendan in different ARDS or sepsis animal models, which somehow compromise the scientific merits of the current study. Also, the ARDS model and the evaluation stage (within 8h after ARDS inducing) may not be an ideal experiment approach for the evaluation of pulmonary inflammation, thereby hindering to draw conclusion of levosimendan's immunoregulatory effect in ARDS.
  3. Figure 2. Why there is big difference in the PaO2/FiO2 ratio among three groups at BLH? Is there statistically significant difference between levosimendan vs vehicle after T0?
  4. Figure 4, pulmonary cytokines levels should be shown. Figures 5, basal line mRNA levels of the tested genes were not shown.
  5. A dot plotting instead of bar graphs will make the readers better evaluate the individual data points across experiments.